# MSI-H Detection by ddPCR in Endoscopic Ultrasound Fine Needle Biopsy (EUS-FNB) from Pancreatic Ductal Adenocarcinoma

**DOI:** 10.3390/ijms252011090

**Published:** 2024-10-15

**Authors:** Maria Assunta Piano, Elisa Boldrin, Lidia Moserle, Nicoletta Salerno, Dalila Fanelli, Giulia Peserico, Maria Raffaella Biasin, Giovanna Magni, Veronica Varano, Giorgia Zalgelli, Vasileios Mourmouras, Antonio Rosato, Antonio Scapinello, Alberto Fantin, Matteo Curtarello

**Affiliations:** 1Immunology and Molecular Oncology Diagnostics Unit, Veneto Institute of Oncology IOV-IRCCS, 35128 Padova, Italy; mariaassunta.piano@iov.veneto.it (M.A.P.); elisa.boldrin@iov.veneto.it (E.B.); lidia.moserle@iov.veneto.it (L.M.); nicoletta.salerno@iov.veneto.it (N.S.); veronica.varano@iov.veneto.it (V.V.); antonio.rosato@iov.veneto.it (A.R.); 2IRCCS Istituto Romagnolo per lo Studio dei Tumori (IRST) “Dino Amadori”, 47014 Meldola, Italy; dalila.fanelli@irst.emr.it; 3Gastroenterology Unit, Veneto Institute of Oncology IOV-IRCCS, 35128 Padova, Italy; giulia.peserico@iov.veneto.it (G.P.); alberto.fantin@iov.veneto.it (A.F.); 4Anatomy and Pathological Histology Unit, Veneto Institute of Oncology IOV-IRCCS, 35128 Padova, Italy; mariaraffaella.biasin@iov.veneto.it (M.R.B.); giorgiazalgelli98@gmail.com (G.Z.); vasileios.mourmouras@iov.veneto.it (V.M.); antonio.scapinello@iov.veneto.it (A.S.); 5Clinical Research Unit, Veneto Institute of Oncology IOV-IRCCS, 35128 Padova, Italy; giovanna.magni@iov.veneto.it; 6Department of Surgery Oncology and Gastroenterology, University of Padova, 35122 Padova, Italy

**Keywords:** microsatellite instability, MSI, pancreatic ductal adenocarcinoma, PDAC, endoscopic ultrasound fine needle biopsy, EUS-FNB, FFPE samples, droplet digital PCR, ddPCR

## Abstract

Pancreatic ductal adenocarcinoma (PDAC) is a highly aggressive disease with limited survival. Curative opportunities are only available for patients with resectable cancer. Palliative chemotherapy is the current standard of care for unresectable tumors. Numerous efforts have been made to investigate new therapeutic strategies for PDAC. Immunotherapy has been found to be effective in treating tumors with high microsatellite instability (MSI-H), including PDAC. The ability of the Endoscopic Ultrasound Fine Needle Biopsy (EUS-FNB) to reliably collect tissue could enhance new personalized treatment by permitting genomic alterations analysis. The aim of this study was to investigate the feasibility of obtaining adequate DNA for molecular analysis from EUS-FNB formalin-fixed-paraffin-embedded (FFPE) specimens. For this purpose, FFPE-DNA obtained from 43 PDAC archival samples was evaluated to verify adequacy in terms of quantity and quality and was tested to evaluate MSI-H status by droplet digital PCR (ddPCR). All samples were suitable for ddPCR analysis. Unlike the 1–2% MSI-H frequency found with traditional techniques, ddPCR detected this phenotype in 16.28% of cases. This study suggests the ddPCR ability to identify MSI-H phenotype, with the possibility of improving the selection of patients who may benefit from immunotherapy and who would be excluded by performing traditional diagnostic methods.

## 1. Introduction

Pancreatic cancer (PC) is a highly malignant disease with a 5-year survival rate of 12%. Its incidence is increasing by about 1% per year [1], and it is expected to become the second leading cause of cancer death in the United States by 2030 [2]. At diagnosis, the majority of patients (75–80%) present a locally advanced or metastatic disease as reviewed in Halbrook et al. [3]. Among PC cases, 96% occur in the exocrine portion, where pancreatic ductal adenocarcinoma (PDAC) accounts for 90% of these [4,5].

The main modality for diagnosing PC is imaging, which commonly involves tri-phasic computed tomography and abdominal magnetic resonance when computed tomography is questionable. Endoscopic Ultrasound (EUS) can be used in some instances for tumor stage assessment and evaluation of vascular involvement. Nevertheless, several studies demonstrated the high sensitivity, specificity and accuracy of the EUS in PDAC detection, especially for small pancreatic masses (0.5–2 cm) [6,7,8,9,10]. EUS can be also used to perform biopsies through EUS-guided fine needle aspiration (EUS-FNA) or fine needle biopsy (EUS-FNB) [11] for diagnostic purposes [12].

Despite the ongoing progress in understanding the molecular mechanisms behind PDAC development, its treatment remains a significant challenge among solid tumors. Surgical resection, followed by adjuvant chemotherapy, represents the primary treatment approach for a small percentage of patients (15–20%) who have a local and resectable tumor mass [13]. Palliative systemic chemotherapy represented by Folfirinox or Gemcitabine and nab-Paclitaxel remains the current standard-of-care for unresectable PDAC [12,14,15]. Nowadays, one of the open questions in PDAC management is defining a proper way of diagnosis, prognosis and predicting therapeutic treatment response.

Targeted therapies, focusing on specific alterations in driver genes involved in tumor progression, have emerged as a promising agnostic treatment approach [16]. A small number of patients are affected by hereditary disease, mainly associated with *BRCA2* pathogenic variants. In the majority of cases (>80%), PC onset is related to somatic mutations that occur sporadically (*KRAS*, *TP53*, *SMAD4* and *p16/CDKN2A*) [17,18].

In the last decade, immunotherapy has been evaluated as a new potential treatment option for PDAC with high microsatellite instability (MSI-H) phenotype, which is a direct consequence of the mismatch repair system deficiency (dMMR) [16,19].

In PDAC, MSI-H/dMMR frequency is estimated in about 1–2% of tumors [20,21,22] and is less common than in other different types of cancers, such as colorectal, endometrial, small bowel and gastric cancers [23,24,25,26,27].

Pembrolizumab’s effectiveness in dMMR tumors was investigated in a phase II trial study in which, among the 86 enrolled patients, 8 were PDAC showing an objective response rate (ORR) of 62% [28]. Moreover, its efficacy was evaluated also by the KEYNOTE-158 trial (NCT02628067) where the 22 treated PDAC patients showed an ORR of 18.2%, a median progression-free survival (PFS) of 2.1 months and a median overall survival (OS) of 4.0 months [29]. Encouraging results were observed in an international multicenter retrospective study of 31 PDAC patients treated with ICI-based therapy, leading to an ORR of 48.4% and a disease control rate (DCR) of 67.7% [30]. Pembrolizumab’s benefit in MSI-H PDAC patients has been also evidenced in several case reports [31,32,33,34]. Following the Keynote-158 study, the FDA and Japanese Authorities approved the agnostic utilization of Pembrolizumab also in the advanced MSI-H/dMMR PDAC [35]. Nevertheless, the European Medicines Agency (EMA) excluded PDAC from the process of treatment approval [36].

MSI-H/dMMR status can be assessed by using different approaches: (i) immunohistochemistry (IHC) staining, which is generally the primary choice as diagnostic tool, and it is used to detect an impaired expression of at least one of the four mismatch repair (MMR) proteins (MLH1, MSH2, MSH6 and PMS2); (ii) molecular methods PCR-based, which test alterations in microsatellite sequences length (used mainly for confirmation in case of doubtful IHC); or (iii) NGS approaches, which analyze the MSI-H phenotype and the Tumor Mutational Burden (TMB) simultaneously [37,38]. Moreover, for scientific purposes, MSI has recently been investigated by using a more sensitive PCR-based technique, such as droplet digital PCR (ddPCR) [39,40,41].

In patients with unresectable PDAC, biopsies assessment for diagnostic evaluation is important in guiding physicians to determine surgical and treatment options. However, the inadequate quantity and quality of extracted DNA could be a limitation in performing molecular analyses.

In the studies conducted by Constantin et al. [42] and Gleeson et al. [43], the MMR status and PD-L1 expression have been successfully assessed by IHC in PDAC EUS-FNB samples. While the studies conducted by Sugimoto et al. [44] and Takagi et al. [45] suggest the feasibility of using the EUS-FNB biopsies for MSI molecular analyses by using multiplex PCR-based methods.

In the era of personalized medicine and with the advent of tumor agnostic therapy, the detection of specific molecular biomarkers using more appropriate techniques is extremely important. The aim of this study was to evaluate the MSI-H status in PDAC EUS-FNB samples by using the highly sensitive ddPCR molecular method, in order to better stratify the patients that could be eligible for important therapeutic opportunities as immunotherapy.

## 2. Results

### 2.1. Clinicopathologic Characteristics of Patients

Forty-three archival formalin-fixed-paraffin-embedded (FFPE) samples of patients diagnosed as PDAC who underwent EUS-FNB were selected. The clinicopathologic characteristics of the patients are shown in Table 1. Briefly, the median age was 69 years (range 51–87), 55.8% were male and 44.2% female. In the majority of them (58%) the tumor was located at the head of the pancreas followed equally by the isthmus (14%), body (14%) and tail (14%). The most representative histological variant is the ductal G1-G2 (79.1%), and in a substantial number of cases (78.6%), PDAC was diagnosed at an advanced stage (III or IV) when the tumor was not resectable.

The median dimension of the tumor mass was 3 cm and ranged between 1.7 and 9 cm. Median levels of the neoplastic markers Carbohydrate Antigen 19-9 (CA 19-9) and Carcinoembryonic Antigen (CEA), evaluated at the diagnosis, were 282.1 U/mL (range, 0.8–35,709 U/mL) and 6.3 ng/mL (range, 1.3–864.0 ng/mL), respectively. Seven patients (16.3%) had intraductal papillary mucinous neoplasm (IPMN) degeneration based on radiological and/or histological evidence. Eleven patients (26.8%) had a family history of cancer.

After the diagnosis, 76.3% of the patients received the standard chemotherapy regimen according to the guideline recommendation. No patient received immunotherapy/ICIs.

### 2.2. DNA Quantity and Quality Assessment

FFPE-DNA has been extracted based on the corresponding hematoxylin/eosin (HE) slide, which was previously digitally scanned and analyzed for the different morphological components.

Considering that PDAC EUS-FNBs are small biopsies, manual macrodissection of neoplastic cells was feasible only in seven samples (16.3%), taking into account the possibility of selecting accurately only tumor component; consequently, DNA extracted from these samples was constituted only by tumor DNA (Table 2). In the remaining 36 samples (83.7%), the whole FFPE section was used for the DNA extraction and, consequently, the obtained DNA represents a mixture of both tumor DNA and normal DNA. In these cases, digital pathology was used to quantify the percentage of tumor area, allowing an estimate of the neoplastic cells’ contribution to the corresponding extracted DNA for each sample. The median tumor area was 19.1% ranging between 5.4% and 72.8% (Table 2).

The median DNA concentration was lower in macrodissected samples (7.22 ng/µL, range: 1.4–16.1 ng/µL) when compared to the not macrodissected samples (11.8 ng/µL, range: 1.4–78.2 ng/µL); on the whole, by considering all samples together, the median DNA concentration obtained was 11.3 ng/µL and ranged between 1.4 and 78.2 ng/µL (Table 2). Furthermore, because DNA degradation is expected to occur during the routine procedure of formalin fixing and paraffin embedding, the DNA Integrity Number (DIN) was evaluated. Median DIN values were comparable between DNA extracted from macrodissected and not macrodissected samples, 3.1 vs. 3.0, respectively, ranging between 1 and 5.9, considering the two groups together. The median length of the fragments in macrodissected samples was 1910 bp vs. 1657.5 in non-macrodissected and, taken together, the median length was 1694 bp (range: 384–10,445 bp) (Table 2).

### 2.3. MSI-H/dMMR Analysis

Once the quantity and the quality of the DNA obtained from the PDAC EUS-FNB were evaluated, we investigated the MSI-H/dMMR status first by using IHC staining, the gold standard method commonly used to detect the dMMR phenotype in other types of tumors. This method is based on the assessment of the four MMR proteins (MLH1, PMS2, MSH2 and MSH6) nuclear expression, and a sample is defined as MSI-H/dMMR by the absence of staining of at least one protein.

Unfortunately, it was not possible to analyze 2 out of 43 samples due to the inadequate staining or due to the unavailability of the FFPE block, and all the remaining 41 samples were identified as microsatellite stable (MSS) because there were no defects in the MMR proteins expression (Table 3).

Once verified, the normal status at the protein level and assuming that IHC is restricted to the staining of the four classical MMR proteins, the next step has been to investigate the MSI-H status also at the molecular level by analyzing changes in the microsatellites’ length in the five mononucleotides repeat markers BAT-25, BAT-26, NR-21, NR-24 and MONO-27 by using the ddPCR approach.

As we have previously demonstrated in Boldrin et al. [46] when compared to other PCR-based methods, ddPCR was able to detect MSI also in small amounts of DNA characterized by a mixture of normal and tumor DNA, with a limit of detection greater than 0.03 ng of tumor DNA as input.

To perform ddPCR we used 5 ng of total DNA as input in the reaction mixture. As mentioned before, in the seven macrodissected samples, this amount represents 100% tumor DNA. On the other hand, in the non-macrodissected samples, the amount of tumor DNA fraction in the entire section was calculated on the basis of the contribution of the tumor area percentage estimated by digital pathology. In these samples, the median amount of tumor DNA was 0.95 ng, with a range between 0.27 ng and 3.64 ng (Table 4 and Figure 1). Altogether, all 43 FFPE-DNA samples extracted from EUS-FNB biopsies proved to be suitable for the ddPCR analysis in terms of the quantity of tumor DNA.

Regarding the ddPCR data interpretation, as detailed in materials and methods, the exact position of the droplet clusters to call each microsatellite (BAT-25, BAT-26, NR-21, NR-24 and MONO-27) as unstable has been determined by using an internal positive control (CTRLpos). Taking into account that, according to national guidelines, the IHC staining for MMR proteins is not routinely performed for PC, for the CTRLpos, we selected a gastric cancer sample. This latter resulted in dMMR/MSI-H according to IHC (PMS2 loss) and showed instability in all five loci by ddPCR (Figure A1). Based on the position of the CTRLpos clusters and of the established cut-off of ≥3 positive droplets to define each microsatellite as unstable, the PDAC sample was defined MSS if none of the loci analyzed showed alteration and MSI-H if at least ≥2 loci were unstable.

A representative image of the IHC staining and the two-dimensional plots for each ddPCR assay from a PDAC sample resulted dMMR/MSI-H in IHC and MSI-H in ddPCR (unstable in BAT-25, BAT-26 and NR-21) and from the CTRLpos is shown in Figure 2.

To avoid false positive results, we also evaluated 19 histologically normal pancreatic tissues as controls. As expected, all normal pancreatic samples resulted in MSS.

However, as shown in Table 3, ddPCR assay has been able to detect MSI-H status in 7 out of 43 samples (16.28%). The IHC staining for five out of these seven MSI-H samples, showing a normal expression of all MMR proteins, is reported in Figure 3.

### 2.4. Association of MSI-H Status with Clinicopathological Features from PDAC Patients

On the basis of the MSI ddPCR results, after categorizing the tumors as MSI-H and MSS, we evaluated the correlation between the MSI-H phenotype and the patient’s clinicopathologic features, as shown in Table 5. No statistically significant differences were found in all the variables that were analyzed. However, it appears that the MSI-H group has a higher frequency of patients with IPMN degeneration than the MSS group (42.9% vs. 11.1%; *p* = 0.0722).

The survival analysis revealed a trend of a better OS for patients with MSI-H phenotype (median OS = 14.6 months; 95% CI, 2.8 months to not reached) compared to the MSS phenotype (median OS = 9.7 months; 95% CI, 5.5 to 13.5 months) (Figure A2 in Appendix A). The curve comparison with the log-rank test revealed no statistically significant differences between the two groups of patients (*p* = 0.3647).

## 3. Discussion

PDAC is an aggressive malignancy with poor survival and represents approximately 90% of all pancreatic cancers [1]. The main factor contributing to the worst prognosis is the delayed diagnosis in the majority of patients, when the tumor is already at an advanced stage due to the asymptomatic progression, precluding them from curative intent surgery.

ICIs have recently been reported to be effective in treating tumors with MSI-H [28,29,30,47]. In PDAC, the incidence rate of MSI-H phenotype ranges between 1 and 2% [20,21,22].

In PC, the use of pancreatic biopsy could be very useful to perform diagnostic typing in order to identify the most effective therapeutic approaches, especially in patients who cannot undergo surgery due to advanced tumor stage. In this study, we evaluated MSI-H status in EUS-FNB samples from 43 PDAC patients. EUS-FNB provides tissue with preserved architecture, which allows for accurate histological diagnosis of PC, and also, it has the potential to provide adequate tissue for IHC staining and molecular profiling, such as DNA or RNA-based marker investigations [11].

Recent studies have demonstrated the feasibility of successfully assessing MSI-H status in PDACs by IHC evaluation of the MMR and PD-L1 proteins expression in EUS-FNB biopsies [42,43,48], and also the possibility of using these biopsies for MSI molecular analysis by PCR-based methods [44,45]. However, since the percentage of tumor DNA isolated from EUS-FNB could be low, we think that PCR-based methods could underestimate the frequency of MSI-H.

To determine MSI-H/dMMR status, we first performed IHC staining, and all samples were found to be MSS. In the majority of cancer types, IHC is the first choice for the diagnosis of MSI-H/dMMR status due to the cost-effective method [49]. However, it has several limitations; indeed, IHC is restricted to detecting the absence/presence of the four traditional MMR proteins (MLH1, PMS2, MSH2 and MSH6), but not the functional consequences of aberrations at the genomic level. Moreover, IHC evaluation and interpretation are often difficult; indeed, one of the IHC’s drawbacks is its propensity to generate false positive/negative results, which may not accurately reflect the MMR system status. This ambiguity can be attributed to multiple factors, including preanalytical, technical, tissue fixation and staining problems [20,49,50]. Moreover, in some cases, the dMMR/MSI-H status may be misclassified by IHC, which may not properly show a defect in the repair mechanism due to mutations in genes coding for the MMR proteins that can maintain normal expression, likely related to retained antigenicity, but make them nonfunctional [51,52]. Furthermore, there could be instances where instability is not caused by aberrations in the four proteins analyzed but rather by other proteins that are involved in the MMR system [53,54].

In order to overcome the IHC limitations, we next investigated MSI status also at the molecular level by using the highly sensitive ddPCR method. To our knowledge, this is the first time that MSI-H status has been performed in EUS-FNB PDAC biopsies by using the ddPCR method.

Due to the characteristics of the pancreatic biopsies, our first step has been to evaluate the quality and the quantity of FFPE-DNA extracted from EUS-FNB biopsies. The DIN was found to have a median value of 3.1, with a range of 1–5.9, in agreement with the literature about DNA extracted from FFPE samples [55]. The average length of the fragments, defined at the main peak by TapeStation analysis, was 1694 bp (range 384–10,445), comparable to a study conducted by Bonnet et al. [56] where the length of the fragments was 1368 bp with DIN, approximately 2.5 for DNA from FFPE samples.

As expected, in our samples, there was a variable amount of total DNA; the median concentration obtained was 11.3 ng/μL with a range of 1.4 to 78.2 ng/μL for a total amount between 35 and 1955 ng in 25 μL of final elution volume.

Considering that for the majority of samples the extracted DNA was a mixture of tumor DNA and normal DNA, the digital pathology allowed us to identify the percentage of tumor area in the FFPE section and then to establish the contribution of the DNA tumor fraction in each sample.

Once the DNA was checked, MSI analyses were performed by ddPCR on all samples. Among the 5 ng of total DNA used as input for each sample, the DNA tumor quantity ranged from 0.27 to 3.64 ng. In all cases, the amount of the estimated tumor DNA was enough for their suitability to perform ddPCR analysis, according to our previously published data where it has been demonstrated that ddPCR was able to detect MSI in an amount greater than 0.03 ng as input of tumor DNA [46].

The ddPCR approach detected MSI-H status in 16.28% of cases, which is greater than the frequency of about 1–2% reported in the literature [20,21,22]. However, over the years, several studies have reported discordant results and substantial differences in the frequencies of MSI-H phenotype in PC patients. These contradictory results can be ascribable to the different approaches used to detect MSI, such as IHC, PCR-based methods and NGS on resected samples from primary tumors or metastases, as reviewed in Lupinacci et al. [57] and Ghidini et al. [58].

Among PCR-based methods, multiplex PCR and real-time PCR are commonly used to detect MSI. However, if compared to ddPCR, their sensitivity differs greatly when the DNA sample is a mixture of normal and tumor DNA, as we have previously demonstrated [46]. Taking into account other possible approaches that could be used to determine MSI status, NGS has shown comparable sensitivity to ddPCR for the detection of genomic alterations [59]. This technique allows millions of DNA fragments to be analyzed simultaneously and can offer the possibility of sequencing at the same time genes encoding many different proteins of the MMR family and also microsatellites. Nevertheless, it is time-consuming, expensive, requires a high DNA quantity and quality [60] and demands strong informatics assistance. Moreover, NGS may not be accurate for sequencing microsatellites because repeated sequences, especially the monomeric ones, may be prone to errors during amplification reactions [61]. An underestimated detection could also depend on an NGS’s low coverage in the regions mapping on microsatellite sequences.

Instead, ddPCR, focusing on some selected microsatellite loci, could be more suitable than NGS in determining the MSI-H phenotype and this reason can explain the greater frequency of MSI-H in our study compared to the literature.

Due to its high sensitivity, ddPCR is currently one of the most effective methods to accurately analyze rare genetic alterations also in small amounts of nucleic acids [59,62] and a useful molecular approach in MSI testing; indeed, it has already been used in different types of cancers, both in solid tumors and in liquid biopsies [39,40,46,53]. In ddPCR technology, the single target sequences are quantified directly by dilution and partitioning the PCR reaction mix into about 20,000 nanodroplets, which can be read and quantified individually, enhancing the probabilities of detecting rare alterations. Target accuracy and detection can be improved using this methodology, saving time and costs.

In our study, ddPCR shows its ability to detect MSI also in specimens with small amounts of tumor DNA mixed with normal DNA, as the nature of the majority of DNA samples obtained from EUS-FNB biopsies. In view of this evidence, we suggest that ddPCR could be a helpful and more sensitive approach to detect MSI-H status if compared to other commonly used methods.

Unfortunately, we have not found a statistically significant correlation with the clinicopathologic characteristics of the patients. Nevertheless, we observed that MSI-H cases are more frequent in IPMN-associated PDAC compared to no-IPMN (42.9% vs. 11.1%; *p* = 0.0722), according to the results of Lupinacci et al. [63] and Hu et al. [20].

One of the main limits of this study is the small size of the patient cohort, which could compromise the achievement of statistical significance in the analysis of correlations. Moreover, considering that this is a retrospective study with only archival FFPE biopsies available, it was not possible to verify if MMR aberrations were of germ-line origin. Indeed, it is well known that approximately 10% of PDAC are related to Lynch syndrome [64,65].

Since MSI-H/dMMR tumors produce numerous neoantigens that can be recognized by the immune system, it could be interesting to investigate the density and spatial distribution of the tumor-infiltrating lymphocytes. Unfortunately, due to the limited amount of biopsy materials, in our study, we did not have the possibility to examine the tumor microenvironment.

In conclusion, this study suggests the great potential of ddPCR to identify MSI-H phenotype in PDAC patients with advanced disease in which the pancreatic biopsy is the only tissue source to perform diagnostic evaluation. The rationale of our findings is that ddPCR, when compared to the traditional diagnostic methods, could offer the opportunity for an improved stratification of patients carrying defects in MMR systems who could benefit from ICIs-based treatment. The introduction of ddPCR technology in the clinical practice to improve the diagnosis of MSI-H status could be very important, considering the recent approval of ICIs as agnostic therapy in the treatment of advanced MSI-H/dMMR PDAC.

## 4. Materials and Methods

### 4.1. Patients

A retrospective cohort of 43 patients diagnosed for PDAC was selected. All patients were referred to the Gastroenterology Unit of the Veneto Institute of Oncology (IOV-IRCCS) and underwent EUS-FNB between May 2019 and July 2020. Inclusion criteria were: (i) ≥18 years of age; (ii) histologically confirmed PDAC diagnosis and its variants (all stages); (iii) the availability of a diagnostic biopsy FFPE tumor block. The patient’s clinical-pathological characteristics were retrieved from the informatic archives.

The study was carried out according to the Code of Ethics of the World Medical Association (Declaration of Helsinki and its later amendments) and had the approval of the Comitato Etico per la Sperimentazione Clinica (CESC) of the Veneto Institute of Oncology (Cod. Int. CESC IOV 2020/84).

### 4.2. Tumor Area Evaluation

The selection and evaluation of the FFPE samples has been performed by an expert pathologist (A.S.). To enrich the neoplastic component, where possible, manual macrodissection was performed. To quantify the tumor area, each slide was digitally scanned by using the Ventana DP200 Slide Scanner (Roche, Monza, Italy), and the QuPath v0.3.0 software analysis (https://qupath.github.io).

### 4.3. Immunohistochemistry

To determine the MMR status (MSI-H/dMMR or MSS) of the pancreatic biopsy specimens, IHC staining for MMR proteins was performed on 3 μm-thick FFPE tissue sections by using the VENTANA MMR IHC panel (Roche, Monza, Italy) containing four primary monoclonal antibodies: mouse anti-MLH1 (M1), mouse anti-PMS2 (A16-4), mouse anti-MSH2 (G219-1129) and rabbit anti-MSH6 (SP93), according to the manufacturer’s instructions. The presence of nuclear staining within the tumor, even if weak, or the absolute absence, denotes the “no loss” or the “loss” of expression of the targeted MMR protein, respectively. Of note, the PMS2 A16-4 clone used in this study is known to provide an inferior performance compared to the other clones, according to NordiQC Immunohistochemistry Quality Control. Furthermore, since the loss of PMS2 is usually associated with a concomitant loss of MLH1 expression, we considered PMS2 staining as convincing, notwithstanding a weak and/or focal positivity, in all the samples showing a strong MLH1 expression by IHC. Therefore, tumors were considered as being MSI-H in the absence of nuclear staining with at least one protein, and MSS in the presence of all four proteins. Each staining pattern was evaluated by an expert pathologist (A.S.).

### 4.4. DNA Extraction

FFPE tumor DNA was extracted from five consecutive 5 µm thick sections of pancreatic biopsies using the QIAamp DNA Micro Kit (Qiagen, Milan, Italy) and eluted in 25 µL of nuclease-free water, while FFPE DNA from normal surgery tissue was extracted from eight consecutive 10 µm thick sections using the QIAamp Mini kit (Qiagen, Milan, Italy). In both cases, extractions were performed according to the manufacturer’s instructions.

DNA was quantified using the Qubit dsDNA High Sensitivity Assay Kit (Thermo Fisher Scientific, Monza, Italy), and the quality was assessed with the Genomic DNA ScreenTape Assay kit using the Agilent TapeStation 4200 (Agilent Technologies, Milan, Italy). DNA Integrity Number (DIN) was automatically calculated by the TapeStation Analysis Software, version 3.2. High DIN value usually corresponds to high DNA quality, whereas low DIN value suggests strongly degraded low-quality DNA. A representative image of the quality of some FFPE-DNA samples is shown in Figure A3 in Appendix A.

### 4.5. ddPCR MSI Molecular Analysis

MSI status assessment was performed by using the Bio-Rad MSI ddPCR test (Bio-Rad, Milan, Italy) according to the manufacturer’s instructions. This approach, is based on the analysis of five microsatellite markers (BAT-25, BAT-26, NR-21, NR-24 and MONO-27) divided into three distinct assays (i) MSI MPX1 Assay 1 (dHsaEXD94202742), (ii) MSI MPX2 Assay 2 (dHsaEXD38375715) and MSI MPX3 Assay 3 (dHsaEDX42742288), and utilizes the competitive drop-off assay by using two different labeled FAM or HEX probes competing for the same target sequence.

PCR mix reaction was performed according to the manufacturer’s protocol and by adding the Uracil-DNA Glycosylase (UDG) (Bio-Rad Cod. Number 12017702) to reduce artifacts in the DNA from formalin-fixed tissues. PCR amplification was performed using the following thermal cycling conditions: enzyme activation 95 °C for 10′ followed by 40 cycles of denaturation at 94 °C for 30″ and 55 °C for 1′ of annealing/extension with a final 98 °C for 10′ for enzyme inactivation. As input, 2.5 ng/well of DNA template was used, and samples were run in duplicate for each assay, resulting in a total of 5 ng of DNA being analyzed. Positive, negative and no-template (nuclease-free water) controls were used for each plate run.

After amplification, the PCR fluorescence products were quantified by the QX200 droplet reader (Bio-Rad) and data were analyzed using the QuantaSoft^TM^ Pro Analysis Software, version 1.0.596 (Bio-Rad, Milan, Italy). Positive, negative and no-template controls have been used as a guide to recognize the exact position of the called microsatellite marker. In order to setup the fluorescence background and exclude false positives, 19 pancreatic normal tissues have been used as controls. Each microsatellite has been considered unstable if at least ≥3 single positive droplets were detected. During the quality check, in case of less than 10,000 droplets per well were generated, the sample was repeated. When applicable, ddPCR experiments were designed, performed, and reported in line with the Digital MIQE Guidelines [66].

The sample was defined as MSS if there was not a variation in the microsatellite’s length at any of the tested loci and MSI-H if an instability was found at ≥2 loci according to the Bethesta guidelines [67]. Thanks to its high sensitivity and specificity, the use of the five mononucleotide repeat markers overcomes the necessity of normal tissue as reference, which is important when there are only small biopsies available.

### 4.6. Statistical Analysis

All categorical variables were compared using either a chi-squared test or a Fisher exact test. Differences in distribution for numerical variables were tested using the non-parametric test Kruskal–Wallis. OS was defined as the time between the diagnosis and the date of death for any cause or until the last access to informatics archives (September 2023). Kaplan-Meier curves were used to illustrate the OS in MSS and MSI-H patients and the differences were tested with the log-rank test. SAS (version 9.4) was used to perform all statistical analyses. A *p*-value < 0.05 was considered statistically significant. The GraphPad Prism software (version 6.0 for Windows, San Diego, San Diego, CA, USA) was used to generate graphs. CorelDRAW 2019 (64-bit) was used to arrange multiple graphs and images.

## Figures and Tables

**Figure 1 ijms-25-11090-f001:**
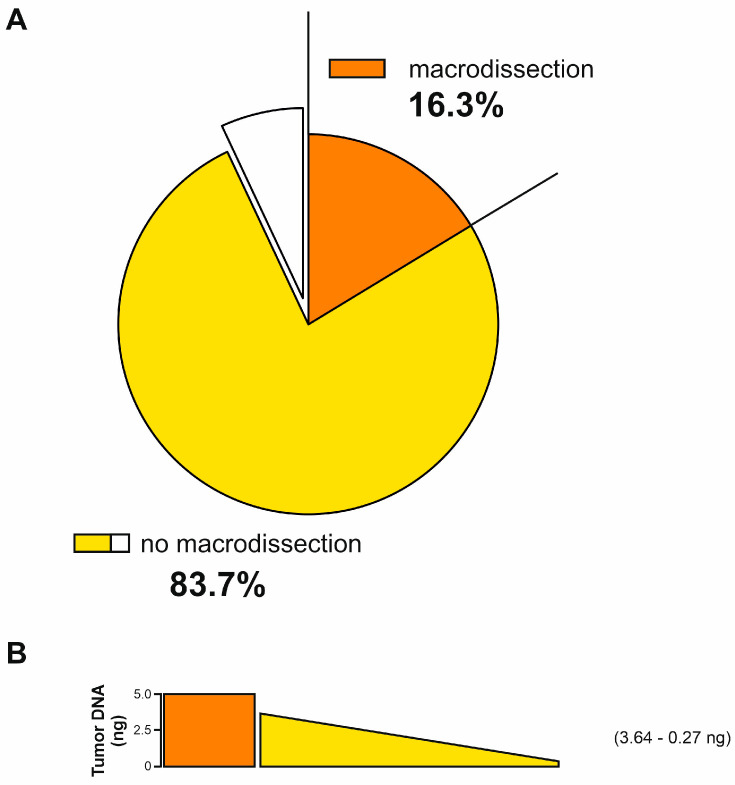
(**A**) Pie chart showing the percentages of formalin-fixed-paraffin-embedded (FFPE) macrodissected samples (16.3%, orange slice) and not macrodissected (83.7%, yellow/white slice). No data were available for 3 samples (white slice). (**B**) Schematic representation of the amount of tumor DNA used as input in droplet digital PCR (ddPCR) analyses. In orange FFPE macrodissected samples, in yellow is represented the range from 3.64 ng to 0.27 ng of tumor DNA used as input in FFPE not macrodissected samples.

**Figure 2 ijms-25-11090-f002:**
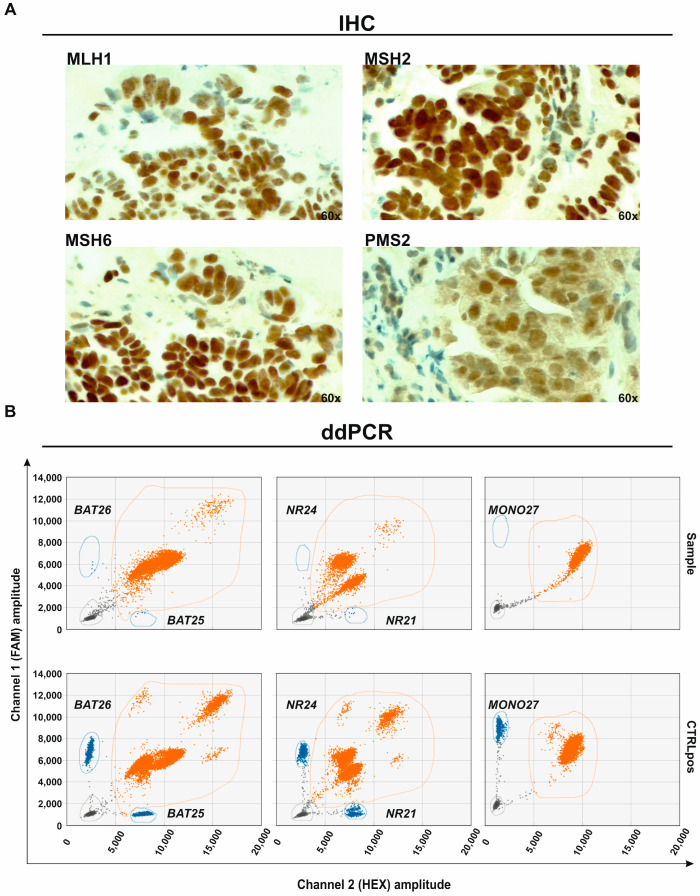
(**A**) Immunohistochemistry (IHC) showing normal expression of the four mismatch repair (MMR) (MLH1, MSH2, MSH6, PMS2) proteins (original magnification 60×) and (**B**) the two-dimensional plots of the five microsatellites marker loci (BAT-25 and BAT-26; NR-21 and NR-24 and MONO-27) analyzed by ddPCR showing instability in three loci (BAT-25, BAT-26 and NR-21) in a representative pancreatic ductal adenocarcinoma (PDAC) FFPE sample. Positive control (CTRLpos) has been used to recognize the exact position of the droplet cluster to call the microsatellite as positive. Orange droplets (orange circle) represent microsatellites with unaltered length, blue droplets (blue circle) represent the microsatellite unstable molecules, and grey droplets (grey circle) represent the ones with the no DNA template.

**Figure 3 ijms-25-11090-f003:**
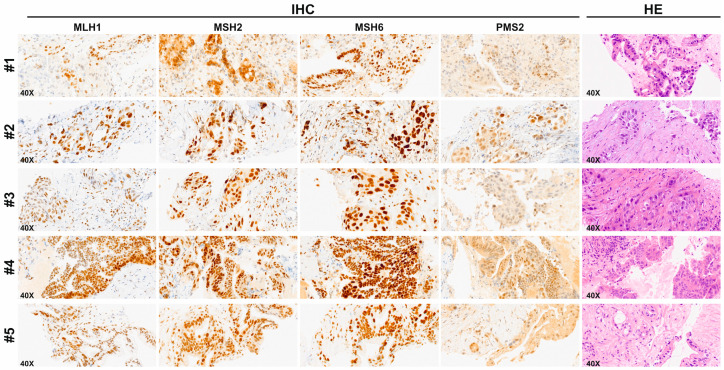
IHC of MMR protein expression and the corresponding hematoxylin/eosin (HE) stain of representative Endoscopic Ultrasound Fine Needle Biopsy (EUS-FNB) specimens of 5 out of the 7 cases resulted in MSI-H according to ddPCR. In all cases, the IHC shows retained nuclear staining of all the four MMR proteins and hence the cases were defined as microsatellite stable (MSS). The HE staining shows a typical PDAC histomorphology (original magnification, 40×).

**Table 1 ijms-25-11090-t001:** Clinicopathologic characteristics of the pancreatic ductal adenocarcinoma (PDAC) patients.

**Patients N (%)**	43 (100)
**Age (years)**	
Median (Q1; Q3)	69 (62; 79)
(Range)	(51–87)
**Gender N (%)**	
Male	24 (55.8%)
Female	19 (44.2%)
**Tumor site N (%)**	
Head	25 (58%)
Isthmus	6 (14%)
Body	6 (14%)
Tail	6 (14%)
**Histologic Variant N (%)**	
Ductal G1-G2	34 (79.1%)
Ductal G3	8 (18.6%)
Ductal with signet-ring component	1 (2.3%)
**Stage**	
I/II	9 (21.4%)
III/IV	33 (78.6%)
Missing	1
**Dimension (cm)**	
Median (Q1; Q3)	3 (2.5; 3.7)
(Range)	(1.7–9.0)
Missing	1
**Neoplastic markers**	
** CA 19-9 (U/mL)**	
Median (Q1; Q3)	282.1 (35.9; 5876.0)
(Range)	(0.8–35,709)
Missing	7
** CEA (ng/mL)**	
Median (Q1; Q3)	6.3 (2.9; 14.5)
(Range)	(1.3–864.0)
Missing	15
**IPMN degeneration**	
Yes	7 (16.3%)
No	36 (83.7%)
**History of cancer**	
Yes	11 (26.8%)
No	30 (73.2%)
Missing	2
**Chemotherapy**	
Yes	29 (76.3%)
No	9 (23.7%)
Missing	5

Q1: first quartile; Q3: third quartile; CA 19-9: Carbohydrate Antigen (CA) 19-9; CEA: Carcinoembryonic Antigen; IPMN: intraductal papillary mucinous neoplasm.

**Table 2 ijms-25-11090-t002:** Quantity and quality of extracted DNA.

	Macrodissection (N; %)	Tumor Area % Median (Range)	DNA ng/µL Median (Range)	DIN Median (Range)	Fragment Length bp Median (Range)
	Yes (7; 16.3%)	100%	7.22 (1.4–16.1)	3.1 (1.0–5.9)	1910 (384–10,445)
	No (36; 83.7%)	19.1% * (5.4–72.8)	11.8 (1.4–78.2)	3.0 (1.1–5.0)	1657.5 (434–2910)
Total	43; 100%	-	11.3 (1.4–78.2)	3.1 (1–5.9)	1694 (384–10,445)

DIN: DNA Integrity Number; * No data were available for 3 samples.

**Table 3 ijms-25-11090-t003:** MSI-H/dMMR status according to IHC and ddPCR.

MSI-H/dMMR Status	IHC N° (%)	ddPCR N° (%)
MSI-H	0	7 (16.28%)
MSS	41 (100%)	36 (83.72%)
Not evaluable	1	0
ND	1	0
Total	43	43

MSI-H: microsatellites instability high; dMMR: mismatch repair deficient; IHC: immunohistochemistry; ddPCR: droplet digital PCR; MSS: microsatellite stable; ND: not done.

**Table 4 ijms-25-11090-t004:** Percentage of tumor area and relative amount of tumor DNA used in ddPCR.

Macrodissection (N; %)	%Tumor Area Median (Range)	ng of Tumor DNA/5 ng of Total DNA as Input in ddPCR Median (Range)
Yes (7; 16.3%)	100%	5 ng
No (36; 83.7%)	19.1% * (5.4–72.8)	0.95 ng * (0.27–3.64)

* No data were available for 3 samples.

**Table 5 ijms-25-11090-t005:** Comparison of the clinicopathologic features between patients with MSI-H and MSS status according to ddPCR analysis.

	MSI-H	MSS	*p*-Value
**N (%)**	7 (16.27%)	36 (83.72%)	
**Age (years)**			
Median (Q1; Q3)	69.0 (56.0; 79.0)	69.5 (63.0; 78.5)	0.4819
(Range)	(51–82)	(55–87)	
**Gender N (%)**			
Male	2 (28.6%)	22 (61.1%)	0.2115
Female	5 (71.4%)	14 (38.9%)	
**Tumor site N (%)**			
Head	4 (57.1%)	21 (58.3%)	0.9999
Isthmus	1 (14.3%)	5 (13.9%)	
Body	1 (14.3%)	5 (13.9%)	
Tail	1 (14.3%)	5 (13.9%)	
**Histologic Variant N (%)**			
Ductal G1-G2	5 (71.4%)	29 (80.5%)	0.7038
Ductal G3	2 (28.6%)	6 (16.7%)	
Ductal with signet-ring component	0	1 (2.8%)	
**Stage N (%)**			
I/II	1 (16.7%)	8 (22.2%)	1
III/IV	5 (83.3%)	28 (77.8%)	
Missing	1	0	
**Dimension (cm)**			
Median (Q1; Q3)	3.0 (2.0; 5.0)	3.0 (2.5; 3.7)	0.68
(Range)	(1.7–8.0)	(2.0–9.0)	
Missing	0	1	
**Neoplastic markers**			
** CA 19-9 (U/mL)**			
Median (Q1; Q3)	406.0 (63.5; 1406)	282.1 (30.0; 8708.0)	0.9799
(Range)	(62.0–2065.0)	(0.8–35,709.0)	
Missing	3	4	
** CEA (ng/mL)**			
Median (Q1; Q3)	1.5 (1.5; 12.9)	6.4 (3.0; 15.4)	0.1692
(Range)	(1.5–12.9)	(1.3–864.0)	
Missing	4	11	
**IPMN degeneration**			0.0722
Yes	3 (42.9%)	4 (11.1%)	
No	4 (57.1%)	32 (88.9%)	
**History of cancer N (%)**			1
Yes	5 (83.3%)	25 (71.4%)	
No	1 (16.7%)	10 (28.6%)	
Missing	1	1	
**Chemotherapy**			1
Yes	4 (80%)	25 (75.8%)	
No	1 (20%)	8 (24.2%)	
Missing	2	3	

## Data Availability

The data supporting the study findings are available from the corresponding author upon reasonable request. The data are not publicly available due to privacy restrictions.

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
