# Peer review of "MSI-H Detection by ddPCR in Endoscopic Ultrasound Fine Needle Biopsy (EUS-FNB) from Pancreatic Ductal Adenocarcinoma"

_ijms, 2024, doi:10.3390/ijms252011090_

Round 1
Reviewer 1 Report
Comments and Suggestions for Authors
The study, entitled " MSI-H detection by ddPCR in Endoscopic Ultrasound Fine Needle Biopsy (EUS-FNB) from Pancreatic Ductal Adenocarcinoma," delineates the utilization of ddPCR for the identification of MSI-H/dMMR tumors.
While the study presents valuable diagnostic approaches, there are a few areas that require further clarification before publication.
Immunohistochemical detection is the method of choice for determining MSI-H/dMMR status. Accordingly, the authors should present positive and negative cases based on immunohistochemistry, together with ddPCR results, to demonstrate the efficacy of the ddPCR method.
The seven cases in which positive MSI-H/dMMR status according to ddPCR is identified are also of interest. As they were not identified based on immunohistochemistry, representative images of these cases, including the immunohistochemistry and HE stain, should be provided so that readers can assess the histomorphology of the cases for themselves.
Furthermore, the tumor microenvironment should be characterized to some extent using CD8 staining to identify potential differences. This will help determine whether there are significant differences between MSI and MSS cases.
Additionally, a comparison of survival between patients who have received systemic therapy with regard to MSI-H/dMMR status is of great interest.
Comments on the Quality of English Language please check the comment
Author Response
"Please see the attachment."

Reviewer 2 Report
Comments and Suggestions for Authors
Maria Assunta Piano and coauthors predict an interesting workflow to identify ddPCR technique for MSI-H detection in Endoscopic Ultrasound Fine Needle Biopsy from Pancreatic Ductal Adenocarcinoma. To validate this result, they used 43 PDAC archival samples from the Gastroenterology Unit of the Veneto Institute of Oncology (IOV-IRCCS) and underwent EUS-FNB between May 2019 and July 2020. Despite the promise, the manuscript has several important flaws that need to be addressed before publication:
The sample number is too meager to give a significant conclusion.
The introduction is to be elaborated. Need to concise it.
In the introduction (Lines 118-120), the Authors wrote several studies but cited only one.
The authors should be adding some H&E images. At least one from the Macro-dissection and another from the remaining samples.
IHC image or heatmap of all four MMR protein is missing.
The authors should explain in the result section, how to investigate MSI-H/dMMR status using IHC.
The representation of five mononucleotide repeat markers BAT-25, BAT-26, NR-21, NR-24 200, and MONO-27 is different in Lines 200 and 223.
Describe the ddPCR result properly. Control positive vs Sample.
References are not uniform.
Some area is highlighted in Ref 53.
Author Response
"Please see the attachment."

Round 2
Reviewer 1 Report
Comments and Suggestions for Authors
Thank you for the detailed reply.
The authors should improve two points that are crucial for the evaluation of the study.
First, better resolved images of the cases shown in Figure 3 should be provided, at least for the review process.
In addition, a higher resolution image of Figure 2 should also be provided, as in Figure 2A the "normal" expression of PMS2 appears very weak compared to the other proteins. Is there a reason for the almost negative expression?
Since the number of CD8-stained cases is very low, it would be highly desirable if the authors could quantify the CD8-positive cells to provide some insight, albeit limited, into the tumor microenvironment. In addition, this comparison should be included in the revised version of the study.
Comments on the Quality of English Languageplease check the report
Reviewer 2 Report
Comments and Suggestions for Authors
Authors response fulfill all drawbacks.
Round 3
Reviewer 1 Report
Comments and Suggestions for Authors
Thank you for the detailed answer.
The authors should describe in the material and methods section, as mentioned in the cover letter, that the PMS2 antibody used shows a comparatively weak staining reaction, as this can otherwise lead to confusion for the reader.
